# SIRT6 regulates Ras-related protein R-Ras2 by lysine defatty-acylation

Xiaoyu Zhang[1], Nicole A Spiegelman[1], Ornella D Nelson[1], Hui Jing[1], Hening Lin[1,2]*

[1]Departmeunt of Chemistry and Chemical Biology, Cornell University, Ithaca, United States; [2]Howard Hughes Medical Institute, Cornell University, Ithaca, United States

**Abstract** The Ras family of GTPases are important in cell signaling and frequently mutated in human tumors. Understanding their regulation is thus important for studying biology and human diseases. Here, we report that a novel posttranslational mechanism, reversible lysine fatty acylation, regulates R-Ras2, a member of the Ras family. SIRT6, a sirtuin with established tumor suppressor function, regulates the lysine fatty acylation of R-Ras2. In mouse embryonic fibroblasts (MEFs), *Sirt6* knockout (KO) increased R-Ras2 lysine fatty acylation. Lysine fatty acylation promotes the plasma membrane localization of R-Ras2 and its interaction with phosphatidylinositol 3-kinase PI3K, leading to activated Akt and increased cell proliferation. Our study establishes lysine fatty acylation as a previously unknown mechanism that regulates the Ras family of GTPases and provides an important mechanism by which SIRT6 functions as a tumor suppressor.

## Introduction

SIRT6 (sirtuin 6) belongs to the Sir2 (silencing information regulator 2) family of nicotinamide adenine dinucleotide (NAD$^+$)-dependent protein lysine deacylases. It plays important roles in a variety of biological processes, including DNA damage and repair (*Kaidi et al., 2010*; *Mao et al., 2011*; *Toiber et al., 2013*), glucose metabolism (*Zhong et al., 2010*), and cell proliferation (*Sebastián et al., 2012*). *Sirt6* knockout (KO) mice display multiple defects and die a few weeks after birth (*Mostoslavsky et al., 2006*). Underlying its biological functions, SIRT6 has multiple enzymatic activities. It can deacetylate histone H3 lysine 9 (Ac-H3K9), lysine 18 (Ac-H3K18), and lysine 56 (Ac-H3K56) (*Michishita et al., 2008*; *Yang et al., 2009*; *Michishita et al., 2009*; *Tasselli et al., 2016*), to suppress target gene expression of several transcription factors, including NF-κB (*Kawahara et al., 2009*), HIF-1α (*Zhong et al., 2010*), c-Jun (*Sundaresan et al., 2012*), and c-Myc (*Sebastián et al., 2012*). SIRT6 has also been reported to be an adenosine diphosphate (ADP)-ribosyltransferase (*Mao et al., 2011*; *Liszt et al., 2005*). We have recently identified SIRT6 as an efficient lysine defatty-acylase that regulates the secretion of tumor necrosis factor (TNFα) (*Jiang et al., 2013*). Mechanistically, lysine fatty acylation promotes TNFα targeting to lysosome and thus decreases its secretion (*Jiang et al., 2016*). However, it remains unclear whether SIRT6 regulates other proteins by defatty-acylation.

The Ras family of proteins plays important roles in numerous biological pathways, including signal transduction, membrane trafficking, nuclear export/import, and cytoskeletal dynamics (*Wennerberg et al., 2005*). Five branches of the Ras superfamily (Ras, Rho, Rab, Arf, and Ran) are classified according to sequence similarity. Ras proteins can exist in two conformational states: a GDP-bound inactive state and a GTP-bound active state. In the GTP-bound state, Ras proteins can recruit effector proteins and turn on specific signaling pathways (*Hancock, 2003*; *Karnoub and Weinberg, 2008*). Protein post-translational modifications (PTMs) play important roles in regulating the Ras family of proteins (*Ahearn et al., 2011*). Ras and Rho families of GTPases are modified by

*For correspondence: hl379@ cornell.edu

Competing interests: The authors declare that no competing interests exist.

**eLife digest** Cancer is one of the leading causes of death worldwide. Proteins that cause and promote cancer are called oncoproteins. Other proteins, called tumor suppressors, counteract the oncoproteins but are frequently inactive or not present in cancer cells.

SIRT6 is a tumor suppressor protein that has been studied in many different types of cancer. In 2013, researchers found that SIRT6 can remove chemical groups known as fatty acyl groups from the lysine residues of proteins. However, it was unclear whether and how this activity of SIRT6 contributes to its role as a tumor suppressor.

Zhang et al. – who are part of the research group who performed the 2013 study – have now compared mouse cells that lack SIRT6 with normal mouse cells to find out which proteins SIRT6 removes fatty acyl groups from. A biochemical technique that makes use of synthetic fatty acids, which get incorporated into the mouse cells, showed that SIRT6 removes fatty acyl groups from a protein called R-Ras2. This protein is part of a large family of oncoproteins.

Zhang et al. discovered that when R-Ras2 is tagged with the fatty acyl group it moves to the cell's membrane and causes the cell to divide more rapidly. Hence, this promotes the growth and spread of cancerous tumors. SIRT6 acts as an eraser, removing the fatty acyl group, and therefore slows down the growth of cancer cells.

Future experiments will aim to find out whether fatty acyl groups also control the activity of other oncoproteins that are similar to R-Ras2. If that is the case, drugs that can regulate the removal of fatty acyl groups from oncoproteins may eventually form new cancer treatment options.

prenylation (farnesylation or geranylgeranylation) on their C-terminal CaaX motif at the cysteine residue. Some proteins from Ras and Rho families, such as H-Ras and N-Ras, contain cysteine palmitoylation as a second lipidation. Both prenylation and palmitoylation serve as important membrane targeting signals (*Ahearn et al., 2011*; *Linder and Deschenes, 2007*). Other proteins from Ras and Rho families, such as K-Ras4B, do not have cysteine palmitoylation and are thought to use a C-terminal polybasic sequence for membrane targeting (*Ahearn et al., 2011*; *Linder and Deschenes, 2007*).

Here, we demonstrate that a Ras family protein, R-Ras2 (also called TC21 because it was cloned from a human teratocarcinoma cDNA library) (*Drivas et al., 1990*), which is able to transform cells and is up-regulated in several human cancers (*Lee et al., 2011*; *Gutierrez-Erlandsson et al., 2013*; *Erdogan et al., 2007*; *Murphy et al., 2002*), is regulated by a novel PTM, lysine fatty acylation. Importantly, SIRT6 is identified as the defatty-acylase of R-Ras2. SIRT6 defatty-acylates R-Ras2 and attenuates its plasma membrane localization, therefore inhibits its ability to activate PI3K signaling pathway and cell proliferation.

## Results

### Defatty-acylation contributes to SIRT6's tumor-suppressor function

SIRT6 has been reported to be a tumor suppressor (*Sebastián et al., 2012*). Decreased SIRT6 expression level is found in several human cancers. Enhanced glycolysis is observed in SIRT6-deficient cells and tumors, which is thought to drive tumor formation in vivo (*Sebastián et al., 2012*). Interestingly, the tumor formation promoted by the loss of SIRT6 is oncogene *HRAS*-independent. However, the exact role of SIRT6 in cancer is still not well understood. In particular, which enzymatic activity is important for the tumor suppression function is not clear.

We have recently identified a single point mutation (G60A) of SIRT6 that maintains the defatty-acylase activity but exhibits no detectable deacetylase activity in cells (*Zhang et al., 2016*). Utilizing this mutant, we first investigated whether the defatty-acylase activity of SIRT6 contributes to tumor suppression and aimed to identify the defatty-acylation substrate protein that is important for this function. We stably expressed SIRT6 wild type (WT) (exhibits both deacetylase and defatty-acylase activities), G60A (exhibits only defatty-acylase activity in cells), or H133Y (exhibits neither activity in cells) into *Sirt6* KO mouse embryonic fibroblasts (MEFs) and tested their effects on cell proliferation.

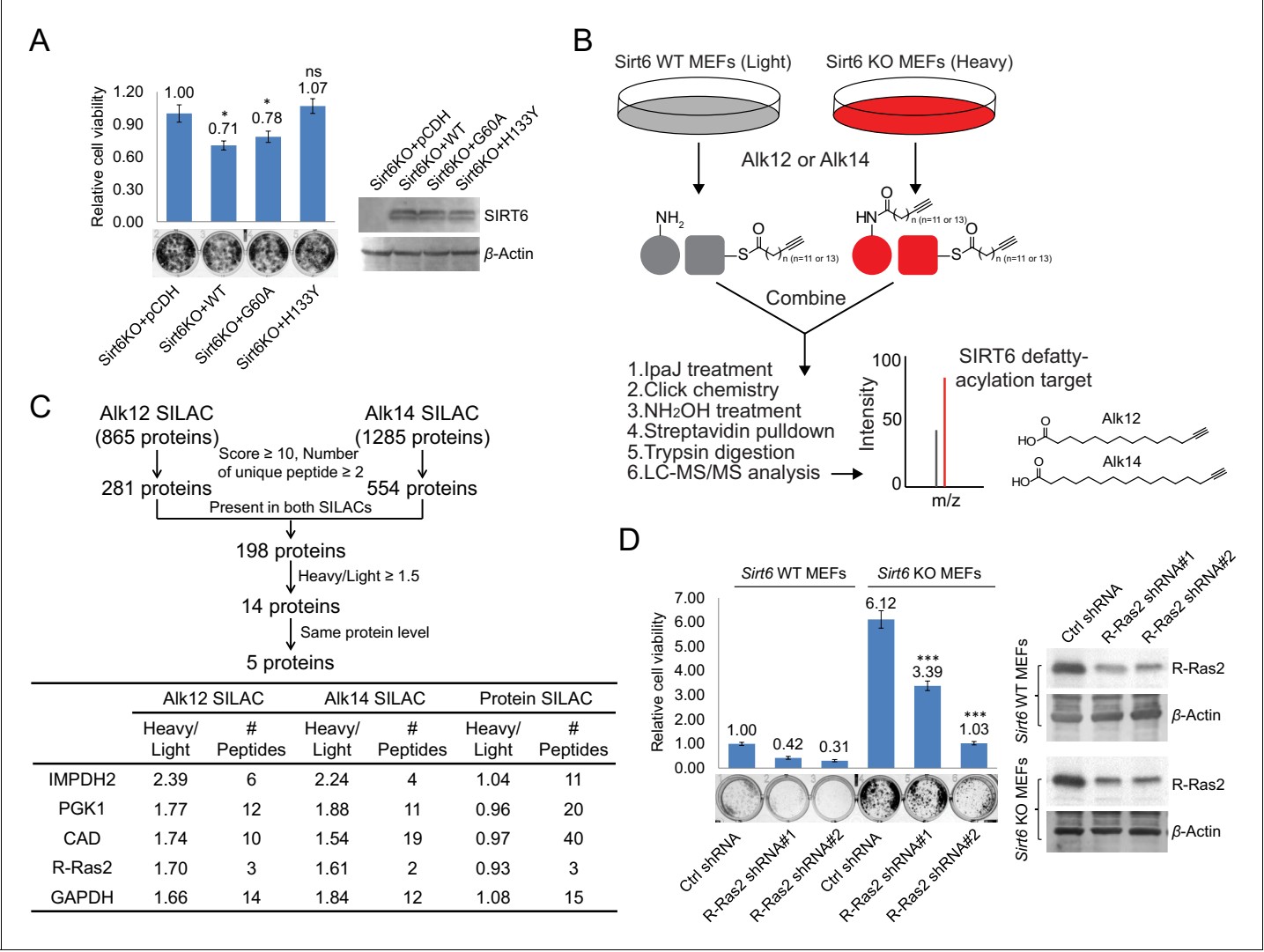

**Figure 1.** Identification of defatty-acylation targets of SIRT6 that contribute to its function in cell proliferation. (**A**) Cell proliferation of *Sirt6* KO MEFs stably expressing pCDH empty vector, SIRT6 WT, G60A, or H133Y. Cell proliferation was assayed and quantified using crystal violet staining. Values with error bars indicate mean ± s.d. of three biological replicates. * indicates p<0.05 and 'ns' indicates no statistical significance when comparing to *Sirt6* KO MEFs with pCDH. (**B**) Schematic overview of the SILAC experiment to identify SIRT6 lysine defatty-acylation targets. (**C**) Data analysis and filter of SILAC results. (**D**) Effects of R-Ras2 knockdown on the proliferation of *Sirt6* WT and KO MEFs. Values with error bars indicate mean ± s.d. of three biological replicates. *** indicates p<0.005 for comparing to *Sirt6* KO MEFs with ctrl shRNA.

The following figure supplement is available for figure 1:

**Figure supplement 1.** Schematic overview of the SILAC experiment to compare the protein abundance in *Sirt6* WT and KO MEFs.

*Sirt6* KO MEFs expressing SIRT6 WT showed decreased cell proliferation compared to those without SIRT6 expression (*Figure 1A*), consistent with the reported role of SIRT6 in suppressing cell proliferation. In contrast, expressing the H133Y mutant had no significant effect on cell proliferation. Interestingly, expression of the G60A mutant, which only exhibits the defatty-acylase activity in cells, also decreased cell proliferation, similar to the expression of SIRT6 WT (*Figure 1A*). The suppression of cell proliferation by SIRT6 WT was only slightly better than that by the G60A mutant. This suggests that although both deacetylase and defatty-acylase activities of SIRT6 contribute, the defatty-acylase activity plays a major role in regulating cell proliferation.

## Identification of SIRT6 defatty-acylation targets

To identify the lysine defatty-acylation targets of SIRT6 that contribute to its tumor suppressing function, we used a quantitative mass spectrometry method, stable isotope labeling with amino acids in cell culture (SILAC), to identify proteins with different lysine fatty acylation levels in *Sirt6* WT and KO MEFs. We used fatty acid analogs to metabolically label fatty acylated proteins and enriched these proteins by incorporating a biotin tag through the copper(I)-catalyzed alkyne-azide cycloaddition (typically called click chemistry) followed by streptavidin pull-down (*Figure 1B*). Two fatty acid analogs were used for cross comparison: Alk12 which better mimics myristic acid and Alk14 which better mimics palmitic acid (*Charron et al., 2009*). When processing the samples, we used IpaJ, a cysteine protease from *Shigella flexneri* that has been reported to hydrolyze the peptide bond after N-myristoylated glycine (*Burnaevskiy et al., 2013*), to reduce the labeling background from N-myristoylation. We also used hydroxylamine ($NH_2OH$) as a nucleophile to remove fatty acylation on cysteine residues.

There were 865 proteins identified in Alk12 SILAC and 1285 proteins identified in Alk14 SILAC (*Supplementary files 1* and *2*). To narrow down the target lists, we filtered the data using four criteria (*Figure 1C*): (1) Protein score $\geq$10, and the number of unique peptide $\geq$2; (2) The target was present in both SILAC experiments, which could decrease the *N*-myristoylation background (proteins present only in Alk12 SILAC) and *S*-palmitoylation background (proteins present only in Alk14 SILAC); (3) Heavy/Light ratio $\geq$1.5, which allowed us to pick proteins that exhibited higher lysine fatty acylation in *Sirt6* KO MEFs than in *Sirt6* WT MEFs; (4) The target had similar protein levels in *Sirt6* WT and KO MEFs by comparing the data from a total protein SILAC experiment (*Figure 1— figure supplement 1*. and *Supplementary file 3*), which was to make sure that proteins with higher Heavy/Light ratios in Alk12 and Alk14 SILAC were not due to increased protein levels. With these criteria, we narrowed down the target lists to five proteins (*Figure 1C*).

Interestingly, a Ras-related small GTPase, R-Ras2, was one of the possible defatty-acylation targets of SIRT6. R-Ras2, like its cousins (H-, N-, and K-Ras), is known to be highly relevant to cancer (*Erdogan et al., 2007*; *Clark et al., 1996*; *Rosário et al., 1999*; *Rong et al., 2002*). We thus hypothesized that the tumor suppressing function of SIRT6 could be through regulating R-Ras2. To test this hypothesis, we knocked down R-Ras2 in *Sirt6* WT and KO MEFs and examined the cell proliferation. Knockdown of R-Ras2 in both *Sirt6* WT and KO MEFs by two different shRNAs significantly decreased cell proliferation (*Figure 1D*), suggesting that R-Ras2 is important for cell proliferation in MEFs.

## R-Ras2 is SIRT6 defatty-acylation target and fatty acylated on C-terminal lysine residues

We then validated whether R-Ras2 had lysine fatty acylation. We first examined R-Ras2 mRNA and proteins levels in *Sirt6* WT and KO MEFs. The results showed that SIRT6 did not affect the transcription and translation of R-Ras2 (*Figure 2A*). We then used the fatty acid analog Alk14 to metabolically label an overexpressed FLAG-tagged R-Ras2. After FLAG immunoprecipitation, we conjugated a fluorescent dye (520-BODIPY-azide) through click chemistry to allow visualization of fatty acylated R-Ras2 by in-gel fluorescence (*Figure 2—figure supplement 1*). We used $NH_2OH$ to remove potential cysteine fatty acylation. Thus, the in-gel fluorescence signal should be mainly attributed to fatty acylation on lysine residues. R-Ras2 exhibited increased lysine ($NH_2OH$-resistant) fatty acylation in *Sirt6* KO MEFs than in WT MEFs (*Figure 2B* and *Figure 2—figure supplement 2A*). We also observed increased R-Ras2 lysine fatty acylation in Human Embryonic Kidney (HEK) 293 T cells after knocking down SIRT6 with two different Sirt6 shRNAs, but not with a control shRNA (*Figure 2C*). We then used an alternative method to detect lysine fatty acylation on overexpressed FLAG-tagged R-Ras2. We treated the cells with Alk14, and then incorporated a biotin tag on fatty acylated proteins (including R-Ras2) through click chemistry. We pulled down the fatty acylated proteins using streptavidin beads and then removed *S*-palmitoylation from the streptavidin beads by $NH_2OH$ treatment. The proteins on beads were then resolved by gel electrophoresis and the level of R-Ras2 was detected by FLAG western blot. This method also revealed more lysine fatty acylation on overexpressed R-Ras2 in *Sirt6* KO MEFs than in *Sirt6* WT MEFs (*Figure 2—figure supplement 2B*).

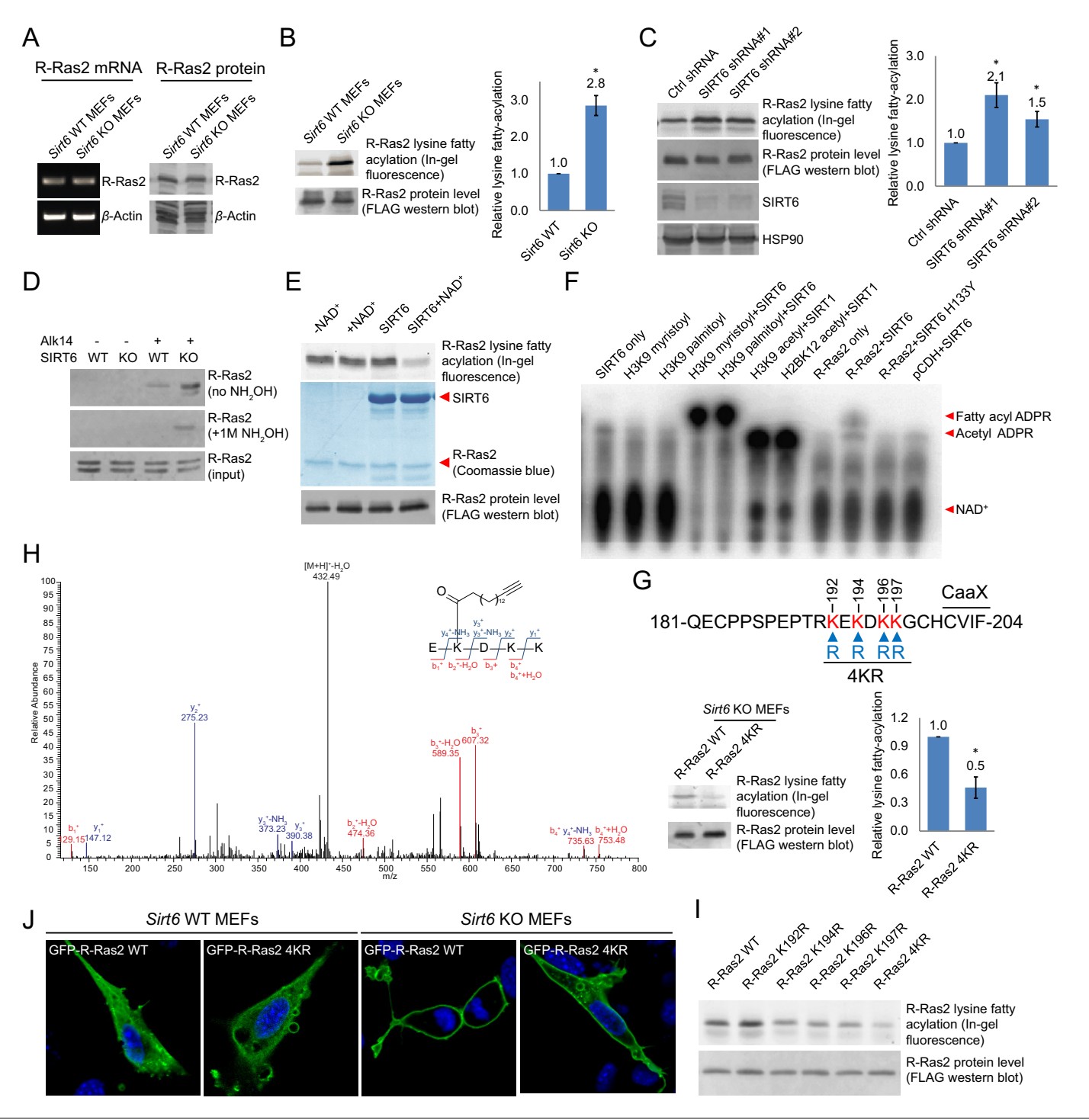

**Figure 2.** Validation of R-Ras2 as the defatty-acylation target of SIRT6. (**A**) mRNA and protein levels of R-Ras2 in *Sirt6* WT and KO MEFs. (**B**) In-gel fluorescence (with NH$_2$OH treatment) showing that R-Ras2 has higher lysine fatty acylation level in *Sirt6* KO MEFs than in *Sirt6* WT MEFs. Right histogram shows the quantification of bands on the fluorescence gel. Values with error bars indicate mean ± s.d. of three biological replicates. * indicates p<0.05. The full fluorescence gel is shown in *Figure 2—figure supplement 2A*. (**C**) Detection of R-Ras2 lysine fatty acylation levels in control and SIRT6 knockown HEK 293 T cells by in-gel fluorescence. Right histogram shows the quantification of bands on the fluorescence gel. Values with error bars indicate mean ± s.d. of three biological replicates. * indicates p<0.05. (**D**) Lysine fatty acylation levels of endogenous R-Ras2 in *Sirt6* WT and KO MEFs. (**E, F**) SIRT6 defatty-acylated R-Ras2 in a NAD$^+$-dependent manner in vitro. In-gel fluorescence was used to detect R-Ras2 lysine fatty acylation (**E**). A $^{32}$P-NAD$^+$ assay was used to detect fatty acyl ADPR product from defatty-acylation reaction. (**F**). (**G**) In-gel fluorescence (with NH$_2$OH
*Figure 2 continued on next page*

*Figure 2 continued*

treatment) showing that mutation of four lysine residues at the C-terminus of R-Ras2 significantly decreased lysine fatty acylation in *Sirt6* KO MEFs. Right histogram shows the quantification of bands on the fluorescence gel. Values with error bars indicate mean ± s.d. of three biological replicates. * indicates p<0.05. The full fluorescence gel including R-Ras2 total fatty acylation levels (without NH$_2$OH treatment) is shown in *Figure 2—figure supplement 2C*. (H) Tandem mass (MS/MS) spectrum of doubly charged Alk14 modified (on K194) R-Ras2 peptide. The b- and y- ions are shown along with the peptide sequence. (I) In-gel fluorescence (with NH$_2$OH treatment) showing that single mutation of four lysine residues at the C-terminus of R-Ras2 did not affect R-Ras2 lysine fatty acylation. (J) Confocal imaging showed that R-Ras2 WT was mainly localized in the plasma membrane in *Sirt6* KO MEFs. R-Ras2 WT in *Sirt6* WT MEFs as well as R-Ras2 4KR in *Sirt6* WT and KO MEFs was localized in both the intracellular vesicles and plasma membrane (n = 5, 5, 5, 6 cells for each sample from left to right, respectively. The images of other cells were shown in *Figure 2—figure supplement 4A*.

The following figure supplements are available for figure 2:

**Figure supplement 1.** Scheme showing the in-gel fluorescence method with Alk14 metabolic labeling to identify R-Ras2 as a lysine fatty acylated protein.

**Figure supplement 2.** Validation of R-Ras2 as the defatty-acylation target of SIRT6.

**Figure supplement 3.** Total ion chromatogram (TIC), extracted ion chromatogram (XIC) and parent MS (MS1) of Ak14 modified R-Ras2 peptide.

**Figure supplement 4.** Lysine fatty acylation targets R-Ras2 to plasma membrane.

Using a similar method, we set out to examine whether endogenous R-Ras2 was regulated by lysine fatty acylation. Endogenous R-Ras2 labeled with Alk14 was conjugated to biotin and pulled down with streptavidin beads. More fatty-acylated endogenous R-Ras2 was pulled down from *Sirt6* KO MEFs than from WT MEFs (*Figure 2D*), suggesting that endogenous R-Ras2 contained more lysine fatty acylation in *Sirt6* KO MEFs.

To confirm that SIRT6 could defatty-acylate R-Ras2 directly, we overexpressed FLAG-tagged R-Ras2 in *Sirt6* KO MEFs, metabolically labeled with Alk14, and then purified R-Ras2 protein from total cell lysates. We then incubated R-Ras2 with SIRT6 in the presence of NAD$^+$ and subsequently performed click chemistry to detect R-Ras2 lysine fatty acylation. In the presence of NAD$^+$, SIRT6 removed most of the lysine fatty acylation signal from R-Ras2 (*Figure 2E*). We also used a previously established $^{32}$P-NAD$^+$ assay (*Du et al., 2011*), in which the newly generated fatty acyl adenosine diphosphate ribose (ADPR) product could be easily detected on thin layer chromatography (TLC) plate, to detect lysine fatty acylation on R-Ras2. SIRT6 WT, but not the catalytic mutant SIRT6 H133Y, generated the fatty acyl ADPR spot in the presence of R-Ras2 isolated from *Sirt6* KO MEFs and NAD$^+$ (*Figure 2F*). These data suggested that R-Ras2 was fatty-acylated on lysine residues and SIRT6 could directly remove the fatty acylation in vitro.

Next, we set out to identify which lysine residues of R-Ras2 were fatty acylated. We noticed that there are four lysine residues in the C-terminal hypervariable region (HVR) of R-Ras2: K192, K194, K196, and K197 (*Figure 2G*). This region is known as a C-terminal polybasic sequence and is generally believed to help anchor Ras proteins to the membrane through electrostatic interaction (*Ahearn et al., 2011*). We suspected that some of these lysine residues might be fatty acylated. We thus mutated these four lysine residues to arginine (the 4KR mutant), which should maintain the positive charge and thus, should not disrupt the membrane association provided by the positive charge. We examined the lysine fatty acylation of R-Ras2 WT and 4KR in *Sirt6* KO MEFs and found that the 4KR mutant significantly decreased the lysine fatty acylation signal (*Figure 2G* and *Figure 2—figure supplement 2C*).

We then utilized mass spectrometry (MS) to directly identify the modification site using FLAG-tagged R-Ras2 purified from Alk14 treated *Sirt6* KO MEFs. A peptide (residue 193–197) carrying Alk14 modification on K194 was detected (*Figure 2H* and *Figure 2—figure supplement 3*). Interestingly, when we mutated each of these four lysine residues to arginine and detected their Alk14 labeling by in-gel fluorescence, the hydroxylamine-resistant labeling of all the single mutants was similar to that of WT (*Figure 2I*). This result suggested that K192, K194, K196, and K197 are likely to be

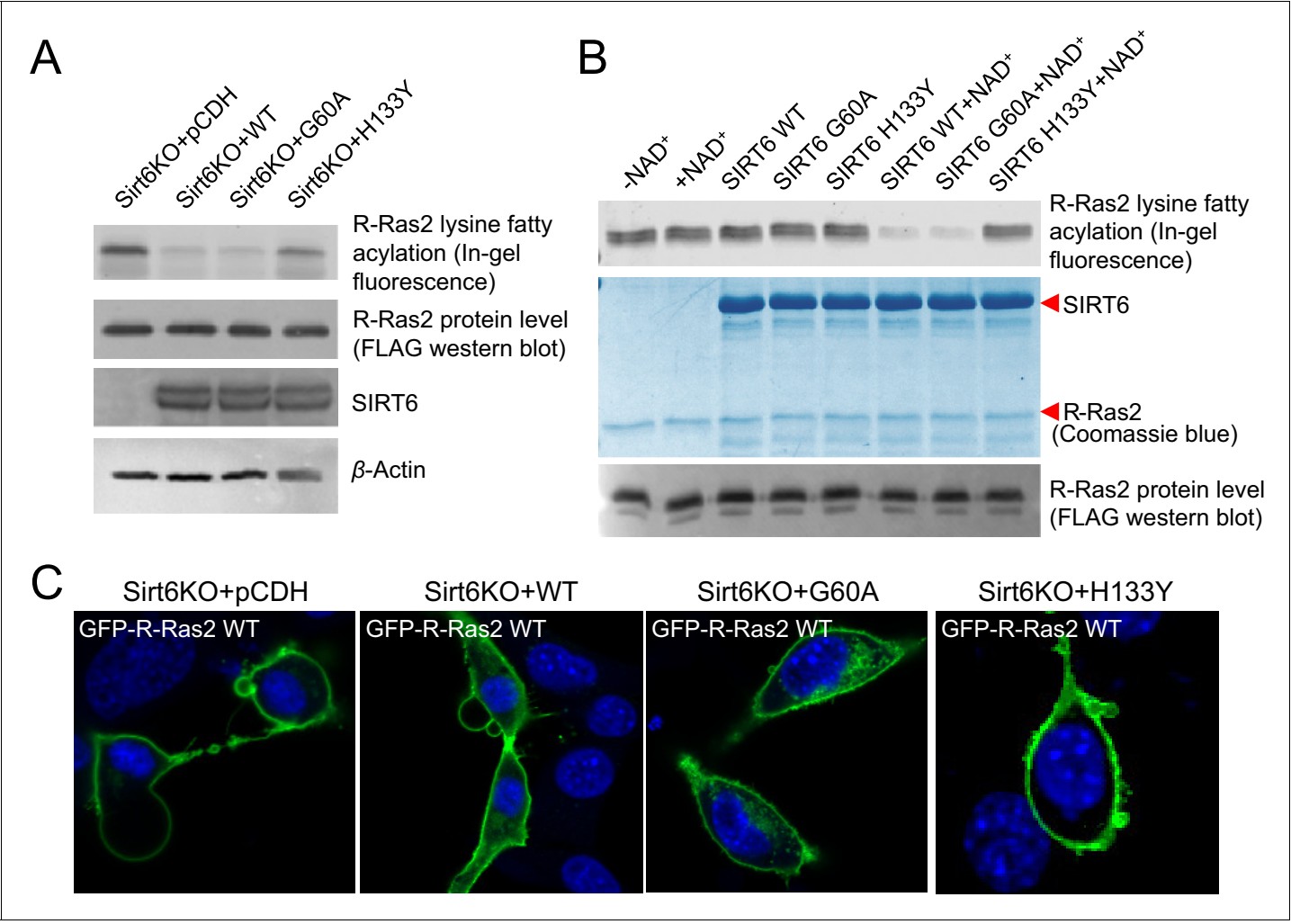

**Figure 3.** SIRT6 defatty-acylase activity is required for regulating R-Ras2 lysine fatty acylation and subcellular localization. (**A**) Detection of R-Ras2 lysine fatty acylation levels in *Sirt6* KO MEFs expressing pCDH empty vector, SIRT6 WT, G60A or H133Y by in-gel fluorescence. (**B**) In-gel fluorescence showed that SIRT6 WT and G60A defatty-acylated R-Ras2 in a NAD$^+$-dependent manner in vitro. (**C**) Confocal imaging showing the localization of GFP-tagged R-Ras2 WT in *Sirt6* KO MEFs expressing pCDH empty vector, SIRT6 WT, G60A or H133Y (n = 5, 6, 5, 5 cells for each sample from left to right, respectively). The images of other cells were shown in *Figure 3—figure supplement 1A*.

The following figure supplement is available for figure 3:

**Figure supplement 1.** The defatty-acylase activity of SIRT6 regulates R-Ras2 subcellular localization.

fatty acylated redundantly, although the MS result implied that K194 is preferentially fatty acylated on WT R-Ras2 protein.

R-Ras2 is known to have palmitoylation on cysteine 199, which is close to the fatty acylated lysine cluster. It is possible that lysine fatty acylation occurs via acyl transfer from the nearby acylated cysteine. To test this possibility, we mutated cysteine 199 to serine (the C199S mutant) and assayed its Alk14 labeling in HEK 293T cells. Alk14 labeling still occurred on R-Ras2 C199S mutant, which was NH$_2$OH resistant (*Figure 2—figure supplement 2D*), suggesting that cysteine fatty acylation of R-Ras2 is not required for the occurrence of its lysine fatty acylation.

### Lysine fatty acylation targets R-Ras2 to plasma membrane

To investigate the function of R-Ras2 lysine fatty acylation, we first examined the subcellular localization of R-Ras2 WT and 4KR in both *Sirt6* WT and KO MEFs by confocal imaging. In *Sirt6* WT MEFs,

both R-Ras2 WT and 4KR were localized in the plasma membrane and intracellular vesicles (*Figure 2J* and *Figure 2—figure supplement 4A*). In contrast, in *Sirt6* KO MEFs, R-Ras2 WT was mainly localized in the plasma membrane, while R-Ras2 4KR was localized in both the plasma membrane and intracellular vesicles (*Figure 2J* and *Figure 2—figure supplement 4A*). Considering that R-Ras2 WT in *Sirt6* KO MEFs has the highest lysine fatty acylation level (*Figure 2B and G*), this data suggests that lysine fatty acylation helps targeting R-Ras2 to the plasma membrane.

In addition, we performed subcellular fractionation of HEK 293T cells overexpressing FLAG-tagged R-Ras2. Under normal conditions, SIRT6 was present and removed most fatty acyl group from R-Ras2 lysine residues, and thus we did not observe obvious difference in plasma membrane and cytosolic localizations between WT R-Ras2 and the 4KR mutant (*Figure 2—figure supplement 4B*). When we treated the cells with palmitic acid (to increase R-Ras2 lysine fatty acylation) or SIRT6 inhibitor TM3 (to inhibit SIRT6 defatty-acylase activity) (*He et al., 2014*), we observed decreased cytosolic localization of WT R-Ras2, but not the 4KR mutant (*Figure 2—figure supplement 4B*). This data also supports that lysine fatty acylation of R-Ras2 targets it to the plasma membrane.

To further confirm that SIRT6 regulates R-Ras2 by defatty-acylation and not deacetylation, we expressed SIRT6 G60A, which exhibits no detectable deacetylase activity in cells, in *Sirt6* KO MEFs and examined the lysine fatty acylation and subcellular localization of R-Ras2. Expressing the G60A mutant in *Sirt6* KO MEFs decreased R-Ras2 lysine fatty acylation to the same level as expressing WT SIRT6 (*Figure 3A*). In vitro, the G60A mutant removed fatty acylation from R-Ras2 similar to WT SIRT6 (*Figure 3B*). Confocal imaging results showed that in *Sirt6* KO MEFs, expression of the G60A mutant promoted R-Ras2 dissociation from the plasma membrane, similar to the expression of WT SIRT6 (*Figure 3C* and *Figure 3—figure supplement 1A*). Subcellular fractionation of endogenous R-Ras2 showed that, similar to WT SIRT6, the G60A mutant decreased the plasma-membrane-localized endogenous R-Ras2 and increased the intracellular vesicle-localized endogenous R-Ras2 (*Figure 3—figure supplement 1B*). These data collectively demonstrated that the defatty-acylase activity of SIRT6 is sufficient to regulate R-Ras2.

## Lysine fatty acylation of R-Ras2 activates PI3K/Akt and promotes cell proliferation

We next investigated how lysine fatty acylation of R-Ras2 affected its downstream signaling effect. There are two well-studied effector pathways of R-Ras2, the PI3K/Akt pathway (*Rong et al., 2002*; *Rosário et al., 2001*) and the Raf/MAPK pathway (*Rosário et al., 1999*). We first examined whether R-Ras2 can activate the Raf/MAPK pathway in MEFs. We used Raf RBD-conjugated argarose beads to pull-down the active GTP-bound form of R-Ras2. The endogenous total Ras (H-Ras, N-Ras and K-Ras), but not the overexpressed or endogenous R-Ras2, was pulled down by Raf RBD (*Figure 4—figure supplement 1*), suggesting that in MEFs, R-Ras2 was unlikely to activate the Raf/MAPK pathway. We then examined the PI3K/Akt pathway by co-immunoprecipitation (co-IP). We overexpressed FLAG-tagged R-Ras2 WT and 4KR in *Sirt6* WT and KO MEFs, immunoprecipitated R-Ras2 and examined p110α, one isoform of the catalytic subunit of PI3K, which has been reported to interact with R-Ras2 (*Rodriguez-Viciana et al., 2004*). WT R-Ras2 in *Sirt6* KO MEFs, which was shown to have the highest level of lysine fatty acylation (*Figure 2B and G*), had more p110α interaction than the WT R-Ras2 in *Sirt6* WT MEFs or the 4KR mutant in either *Sirt6* WT or KO MEFs (*Figure 4A*). This data suggested that lysine fatty acylation of R-Ras2 promotes its interaction with p110α.

Then, we examined whether the interaction of R-Ras2 with p110α could activate its downstream kinase Akt. *Sirt6* KO MEFs showed higher phosphorylated Akt on Thr308 (p-Akt Thr308), but not on Ser473 (p-Akt Ser473), when compared to *Sirt6* WT MEFs (*Figure 4B*). Knockdown of R-Ras2 in *Sirt6* WT MEFs did not affect p-Akt Thr308, whereas knockdown of R-Ras2 in *Sirt6* KO MEFs decreased p-Akt Thr308 to the same level as that in *Sirt6* WT MEFs (*Figure 4B* and *Figure 4—figure supplement 2A*). We further found that expressing WT SIRT6 or the G60A mutant in *Sirt6* KO MEFs decreased p-Akt Thr308 levels (*Figure 4C* and *Figure 4—figure supplement 2B*), suggesting that the defatty-acylation activity of SIRT6 is important for regulating p-Akt Thr308. All the data combined suggested that SIRT6 defatty-acylates R-Ras2 and attenuates its interaction with p110α, resulting in decreased Akt phosphorylation on Thr308.

Finally, to confirm that lysine fatty acylation of R-Ras2 is important for its role in promoting cell proliferation, we overexpressed R-Ras2 WT and 4KR in *Sirt6* WT and KO MEFs and measured the cell proliferation. Expression of R-Ras2 WT in *Sirt6* KO MEFs significantly increased cell proliferation

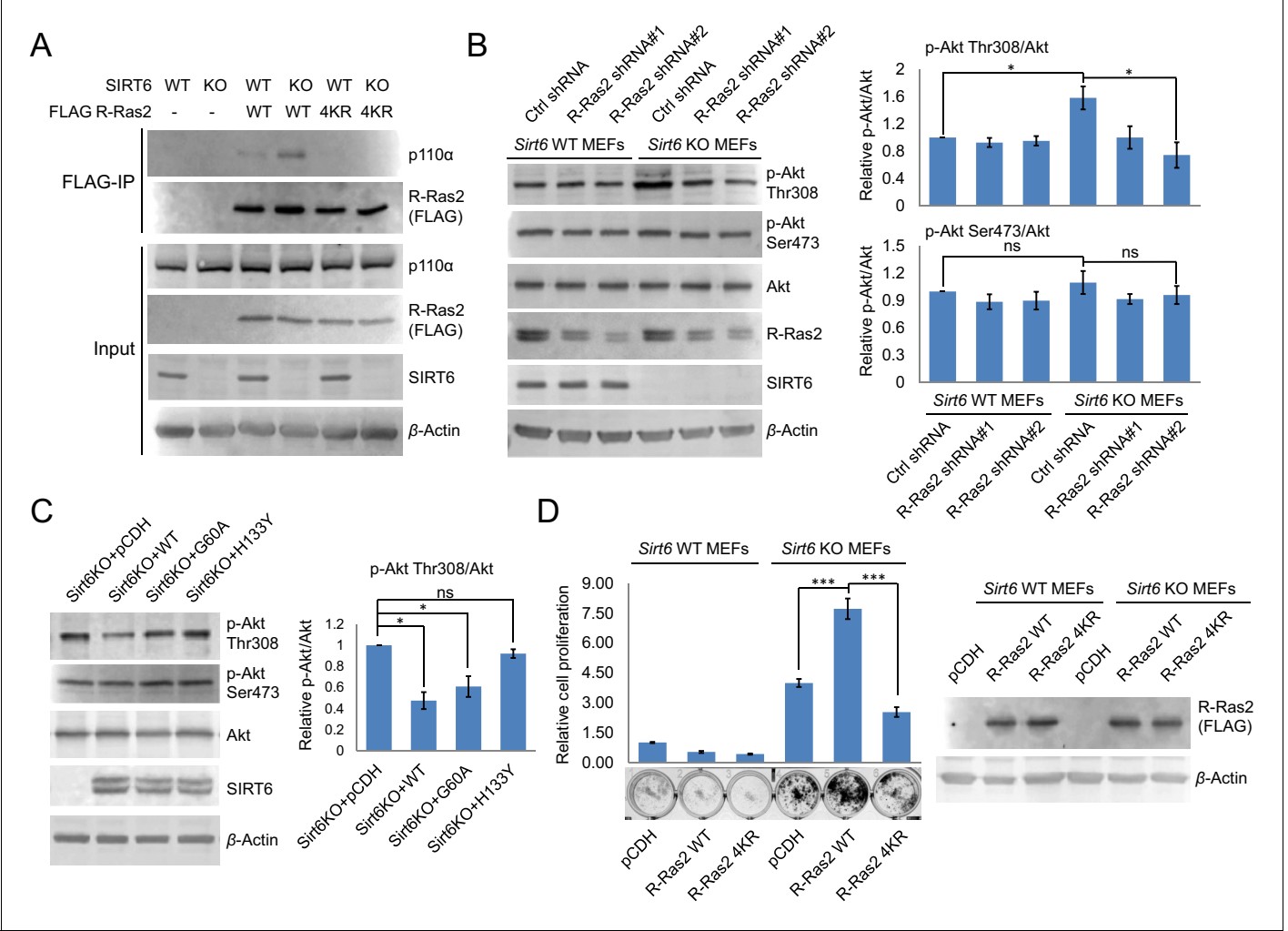

**Figure 4.** Lysine fatty acylation of R-Ras2 activates PI3K-Akt pathway and promotes cell proliferation in MEFs. (**A**) Co-IP experiment showed that R-Ras2 WT in *Sirt6* KO MEFs had more p110α interaction than R-Ras2 WT in *Sirt6* WT MEFs and R-Ras2 4KR in *Sirt6* WT or KO MEFs. (**B**) Knockdown of R-Ras2 in *Sirt6* KO MEFs decreased p-Akt Thr308, but not p-Akt Ser473, to a level similar to that in *Sirt6* WT MEFs. Right histogram shows the quantification of bands on the Western blot membrane. Values with error bars indicate mean ± s.d. of three biological replicates. * indicates p<0.05. The images of the other western blots used for quantification are shown in *Figure 4—figure supplement 2A*. (**C**) Expressing SIRT6 WT or SIRT6 G60A in *Sirt6* KO MEFs decreased p-Akt Thr308, but not p-Akt Ser473. Right histogram shows the quantification of bands on the western blot membrane. Values with error bars indicate mean ± s.d. of three biological replicates. * indicates p<0.05. The images of other western blots for quantification are shown in *Figure 4—figure supplement 2B*. (**D**) R-Ras2 WT but not R-Ras2 4KR promoted cell proliferation in *Sirt6* KO MEFs. Values with error bars indicate mean ± s.d. of three biological replicates. *** indicates p<0.005.

The following figure supplements are available for figure 4:

**Figure supplement 1.** Raf-RBD binding assay showed that both overexpressed and endogenous R-Ras2 did not bind Raf RBD-conjugated argarose beads.

**Figure supplement 2.** The images of other western blots for quantification.

compared with vector control (*Figure 4D*), while expressing R-Ras2 4KR in *Sirt6* KO MEFs did not increase cell proliferation (*Figure 4D*). In *Sirt6* WT MEFs, neither overexpressing WT R-Ras2 nor the 4KR mutant increased cell proliferation compared with vector control (*Figure 4D*). These data suggested that lysine fatty acylation of R-Ras2 is important for promoting MEF cell proliferation.

## Discussion

In this study, we found that the defatty-acylase activity of SIRT6 is important for regulating cell proliferation. Using a proteomics approach, we identified a small GTPase, R-Ras2, as a defatty-acylation target of SIRT6. R-Ras2 has a higher lysine fatty acylation level in *Sirt6* KO MEFs than that in *Sirt6* WT MEFs. Lysine fatty acylation targets R-Ras2 to the plasma membrane, facilitates its interaction with p110α, and activates the Akt signaling pathway. This is important for the proliferative phenotype of *Sirt6* KO MEFs because when we knocked down R-Ras2 or decreased its lysine fatty acylation in *Sirt6* KO MEFs, the p-Akt Thr308 level and cell proliferation became similar to those in *Sirt6* WT MEFs.

Previously, the function of SIRT6 was mainly explained through transcriptional regulation, which typically occurs at the end of signal transduction pathways. Now we showed that SIRT6 can also function at the top of the signal transduction pathway by regulating a GTPase in the Ras family. SIRT6 is known to be a tumor suppressor and down-regulated in many different human cancers (*Sebastián et al., 2012*). Our study suggests that *Sirt6* KO increases lysine fatty acylation of R-Ras2 and activates the PI3K-Akt pathway, leading to higher cell proliferation. Thus, the defatty-acylation and suppression of R-Ras2 is a major contributor to the tumor suppressor role of SIRT6.

For the Ras proteins that do not have palmitoylated cysteine (such as K-Ras4B), they are thought to use polybasic sequences for membrane targeting (*Hancock, 2003*; *Cox et al., 2014*). Interestingly, in addition to palmitoylation on Cys199 and farnesylation on Cys201, R-Ras2 contains several lysine residues at the C-terminus (Lys192, 194, 196, 197). One question is, if these lysine residues also help to target R-Ras2 to membranes via electrostatics, why cells use lysine instead of arginine? Our study has shed lights onto this question and identified a novel regulatory mechanism for this family of proteins, which may help to develop new strategies to pharmacologically target R-Ras2 to treat human diseases. In a recent paper, the lysine residues at the C-terminal of K-Ras4B have been shown to play important roles for phospholipid binding and K-Ras4B signal output (*Zhou et al., 2017*). The lysine to arginine mutant had distinct lipid-binding capacity, suggesting that simple electrostatic interaction may not be the mechanism how polybasic sequences target to the membrane. It is possible that the lysine residues at the C-terminal of K-Ras4B are similarly regulated by fatty acylation, which could explain the distinct lipid-binding capacity of the lysine-to-arginine mutant. However, we could not completely rule out that the lysine residues at the C-terminal of R-Ras2 may also function in phospholipid binding and thus affect its signaling output, in addition to being regulated by lysine fatty acylation and SIRT6.

Many Ras family of small GTPases contain polybasic sequences at the C-termini. It is possible that lysine fatty acylation also occur on these GTPases and regulate their localization and activities. If so, study how these GTPases are regulated by lysine fatty acylation and whether sirtuins also serve as deacylases for these proteins is an ongoing direction in our laboratory.

Our study also expands the biological significance of protein lysine fatty acylation. Protein lysine fatty acylation was first reported in 1992 (*Stevenson et al., 1992*). Since then, only a few proteins were known to have this modification and whether this PTM has important biological functions or not was not known. We previously identified that lysine fatty acylation regulates protein secretion (e.g. TNFα and ribosomal proteins) (*Jiang et al., 2013*; *Zhang et al., 2016*). The finding that lysine fatty acylation regulates the Ras family of proteins suggest that this PTM may have very broad and important biological functions.

## Materials and methods

### Reagents

Anti-FLAG affinity gel (#A2220, RRID: AB_10063035) and anti-FLAG antibody conjugated with horseradish peroxidase (#A8592, RRID: AB_439702) were purchased from Sigma. Human/mouse SIRT6 (#12486), p110α (#4249, RRID: AB_2165248), p-Akt Thr308 (#13038, RRID: AB_2629447), p-Akt Ser473 (#4060, RRID: AB_2315049), Akt (#4691, RRID: AB_915783), HSP90 (#4877, RRID: AB_2121214) and Na,K-ATPase (#3010, RRID: AB_2060983) antibodies were purchased from Cell Signaling Technology. β-Actin (sc-4777) and GAPDH (sc-20357, RRID:AB_641107) antibodies were purchased from Santa Cruz Biotechnology. R-Ras2 antibody (#H00022800-M01, RRID:AB_547895) was purchased from Novus Biologicals. $^{32}$P-NAD$^+$ was purchased from PerkinElmer. 3X FLAG peptide,

Azide-PEG3-biotin, Tris[(1-benzyl-1H-1,2,3-triazol-4-yl)methyl]amine (TBTA), Tris(2-carboxyethyl) phosphine (TCEP), hydroxylamine, NAD$^+$, and protease inhibitor cocktail were purchased from Sigma. Sequencing grade modified trypsin and FuGene six transfection reagent were purchased from Promega (Madison, WI). ECL plus western blotting detection reagent and Streptavidin agarose beads were purchased from Thermo Scientific Pierce (Rockford, IL). Sep-Pak C18 cartridge and poly-ester-backed silica plate were purchased from Waters (Milford, MA). 520-BODIPY azide was purchased from Active Motif (Carlsbad, CA). Ras assay kit (Raf-1 RBD, agarose) was purchased from EMD Millipore (Temecula, CA). R-Ras2 and SIRT6 shRNA lentiviral plasmids (pLKO.1-puro vector) were purchased from Sigma, the sequence of shRNA was: R-Ras2 shRNA#1: TRCN0000306170 (ccggtaagagtcccttgaggtttagctcgagctaaacctcaagggactcttatttttg); R-Ras2 shRNA#2: TRCN0000077748 (ccggcgctagatattgactgttatactcgagtataacagtcaatatctagcgttttttg); SIRT6 shRNA#1: TRCN0000378253 (ccggcagtacgtccgagacacagtcctcgaggactgtgctcggacgtactgttttttg); SIRT6 shRNA#2: TRCN0000232528 (ccgggaagaatgtgccaagtgtaagctcgagcttacacttggcacattcttcttttttg). Alk12 and Alk14 were synthesized according to reported procedures (*Charron et al., 2009*). Plasmid of IpaJ in pET2-8b vector was a kind gift from Prof. Neal M. Alto at Department of Microbiology, University of Texas Southwestern Medical Center. IpaJ was purified according to reported procedures (*Burnaevskiy et al., 2013*).

## Cell culture

*Sirt6* WT and knockout (KO) MEFs were kindly provided by Prof. Raul Mostoslavsky at Massachusetts General Hospital Cancer Center, Harvard Medical School, which were prepared and authenticated by the authors as described previously (*Sebastián et al., 2012*). Human Embryonic Kidney (HEK) 293T cells were purchased from ATCC (RRID: CVCL_0063). All the cell lines have been tested for mycoplasma contamination by PCR-based mycoplasma detection kit (Sigma, MP0025) and showed no contamination. *Sirt6* WT, *Sirt6* KO MEFs and *Sirt6* KO MEFs expressing SIRT6 WT, G60A or H133Y were cultured in Dulbecco's Modified Eagle Medium (DMEM) with 10% fetal bovine serum (FBS). HEK 293T cells were cultured in DMEM medium with 10% FBS.

## Cloning, expression, and purification of SIRT6 WT, G60A and H133Y from *Escherichia coli*

Human SIRT6 was inserted into pET-28a vector. SIRT6 G60A and H133Y mutants were made by QuikChange. The plasmids of SIRT6 WT and mutants were transformed into *E. coli* BL21 (DE3) cells. The proteins were purified according to reported procedures (*Jiang et al., 2013*).

## Cloning, expression, and purification of R-Ras2 WT and 4KR from *Sirt6* KO MEFs and HEK 293T cells

Human R-Ras2 was inserted into lentiviral vector (pCDH-CMV-MCS-EF1-Puro) with N-FLAG tag. R-Ras2 4KR mutant was made by QuikChange. R-Ras2 lentivirus was generated by co-transfection of R-Ras2, pCMV-dR8.2, and pMD2.G into HEK 293T cells. After transfection for 48 hr, the medium was collected and used for infecting *Sirt6* KO MEFs or HEK 293T cells. To obtain the R-Ras2 stable overexpressed cells, the cells were treated by 1.5 mg/mL of puromycin 48 hr after infection and cultured for 1 week while passing cells every 2–3 days. To purify R-Ras2 from *Sirt6* KO MEFs or HEK 293T cells, the cells were collected at 500 g for 5 min and then lysed in Nonidet P-40 lysis buffer (25 mM Tris-HCl pH 7.4, 150 mM NaCl, 10% glycerol and 1% Nonidet P-40) with protease inhibitor cocktail (1:100 dilution). Anti-FLAG affinity gel was used to enrich the R-Ras2 protein from total cell lysates. Then R-Ras2 was eluted and purified by 3X FLAG peptide following the manual.

## Stable overexpression of SIRT6 WT, G60A or H133Y in *Sirt6* KO MEFs

Human SIRT6 WT was inserted into pCDH-CMV-MCS-EF1-Puro vector without tag. SIRT6 G60A and H133Y mutants were made by QuikChange. SIRT6 lentivirus was generated by co-transfection of SIRT6, pCMV-dR8.2, and pMD2.G into HEK 293T cells. After transfection for 48 hr, the medium was collected and used for infecting *Sirt6* KO MEFs. *Sirt6* KO MEFs with stable expressed SIRT6 WT, G60A or H133Y was selected by 1.5 mg/mL of puromycin. Empty pCDH vector was used as the negative control.

## Labeling of R-Ras2 in MEFs with Alk14 and detection of lysine fatty acylation on R-Ras2 by in-gel fluorescence

R-Ras2 WT or 4KR was transfected into MEFs or HEK 293T cells by FuGene six transfection reagent. After 24 hr, the cells were treated with 50 μM of Alk14 for 6 hr. The cells were collected at 500 g for 5 min and then lysed in Nonidet P-40 lysis buffer (25 mM Tris-HCl pH 7.4, 150 mM NaCl, 10% glycerol, and 1% Nonidet P-40) with protease inhibitor cocktail. The total lysate was incubated with anti-FLAG affinity gel at 4°C for 1 hr. The affinity gel was then washed three times by immunoprecipitation (IP) washing buffer (25 mM Tris-HCl pH 7.4, 150 mM NaCl and 0.2% Nonidet P-40) and then re-suspended in 18 μL of IP washing buffer. The click chemistry reaction was performed by adding the following reagents: 520-BODIPY azide (0.8 μL of 1.5 mM solution in DMF), TBTA (1.2 μL of 10 mM solution in DMF), $CuSO_4$ (1 μL of 40 mM solution in $H_2O$) and TCEP (1 μL of 40 mM solution in $H_2O$). The reaction was allowed to proceed at room temperature for 45 min. Then, the SDS loading buffer was added and heated at 95°C for 10 min. After centrifugation at 15,000 g for 2 min, the supernatant was collected and treated with 300 mM hydroxylamine at 95°C for 7 min. The samples were resolved by 12% SDS-PAGE. In-gel fluorescence signal was recorded by Typhoon 9400 Variable Mode Imager (GE Healthcare Life Sciences).

## Detection of lysine fatty acylation of endogenous R-Ras2 in MEFs

*Sirt6* WT and KO MEFs were treated with 50 μM Alk14 for 6 hr. Cells were collected and lysed by Nonidet P-40 lysis buffer using the same method descried above. The total lysates were subjected to click chemistry reaction by adding the following reagents: Azide-PEG3-biotin (final concentration was 100 μM), TBTA (final concentration was 0.5 mM), $CuSO_4$ (final concentration was 1 mM), and TCEP (final concentration was 1 mM). The reaction was allowed to proceed at room temperature for 45 min, and then the total proteins were precipitated by methanol/chloroform (2.5/1) and washed by ice-cold methanol. The protein pellets were re-solubilized in 1.5% SDS, 1% Brij97, 100 mM NaCl and 50 mM triethanolamine. Streptavidin agarose beads were added and incubated at room temperature for 1 hr. After washing the beads three times by 0.2% SDS in PBS buffer, the beads were treated with 1 M hydroxylamine (pH 7.4) at room temperature for 1 hr. Then, the beads were washed three times with 0.2% SDS in PBS buffer. The SDS loading buffer was added to the beads, heated at 95°C for 10 min, and then used for Western blot.

## Defatty-acylation of R-Ras2 by SIRT6 in vitro

R-Ras2 WT was transfected into *Sirt6* KO MEFs treated with 50 μM Alk14 for 6 hr and purified using the method described above. The in vitro assay was proceeded in the assay buffer (50 mM Tris-HCl pH 8.0, 100 mM NaCl, 2 mM $MgCl_2$, 1 mM DTT, 1 mM $NAD^+$) with 15 μM of SIRT6 and incubated at 37°C for 2 hr. Proteins were precipitated by methanol/chloroform (2.5/1) and washed by ice-cold methanol. The protein pellets were re-solubilized in 4% SDS, 150 mM NaCl and 50 mM triethanolamine. Then click chemistry reaction and in-gel fluorescence were carried out as described above.

## $^{32}P$-$NAD^+$ assay

R-Ras2 WT was transfected into *Sirt6* KO MEFs and purified by FLAG affinity gel. Purified R-Ras2 on the FLAG affinity gel was used for $^{32}P$-$NAD^+$ assay. 10 μL of reaction buffer containing 50 mM Tris pH 8.0, 150 mM NaCl, 1 mM DTT, 5 μM SIRT6 WT or H133Y, and 0.1 μCi of $^{32}P$-NAD was mixed with R-Ras2 protein. The reaction was allowed to proceed at 37°C for 2 hr. 2 μL of the reaction mixture was spotted onto the polyester-backed silica plate. The plate was developed in 30:70 (v/v) 1 M ammonium bicarbonate/95% ethanol. Then, the plate was exposed in the phosphor imaging screen (GE Healthcare, Piscataway, NJ) for 4 hr. The signal was detected using Typhoon 9400 Variable Mode Imager. H3K9 myristoyl and H3K9 palmitoyl peptides were incubated with SIRT6 in the same reaction buffer as positive control for fatty acyl ADPR. H2BK12 acetyl peptide was incubated with SIRT1 in the same reaction buffer as positive control for acetyl ADPR. All the peptides were synthesized according to the reported procedures (*Zhu et al., 2012*).

### Western blot

The proteins were resolved by 12% SDS-PAGE and transferred to polyvinylidene fluoride (PVDF) membrane. The membrane was incubated with 5% bovine serum albumin (BSA) in TPBS buffer (0.1%

Tween-20 in PBS solution) at room temperature for 60 min. Then, the antibody was diluted with fresh 5% BSA in TPBS buffer and incubated with the membrane for different time points according to the manual. After washing three times by TPBS buffer, the secondary antibody was diluted with fresh 5% BSA in TPBS buffer and then incubated with the membrane at room temperature for 1 hr. The chemiluminescence signal in membrane was recorded after developing in ECL plus western blotting detection reagents using Typhoon 9400 Variable Mode Imager.

## Crystal violet cell proliferation assay

Cells were seeded in 12-well plates (5000 cells/well) or 24-well plates (2500 cells/well), and then maintained in DMEM medium with 10% FBS for 5 days. After washing twice with ice-cold PBS, cells were fixed by ice-cold methanol for 10 min. Then, methanol was removed and crystal violet (0.2% in 2% ethanol solution) was added and incubated for 5 min. Cells were then washed with water until all excess dye was removed. Crystal violet dye that remained with the cells was solubilized by 0.5% SDS in 50% ethanol solution. The absorption of crystal violet was measured at 550 nm.

## SILAC

*Sirt6* KO MEFs were cultured in DMEM medium with [$^{13}C_6$, $^{15}N_2$]-L-lysine and [$^{13}C_6$, $^{15}N_4$]-L-arginine for five generations. *Sirt6* WT MEFs were cultured in normal DMEM medium for five generations. The cells were treated with 50 µM Alk12 or Alk14 for 6 hr. After quantifying the concentration of total proteins by Bradford assay, 2.5 mg of total proteins from each sample were mixed. IpaJ was added (final concentration was 150 µg/mL) and incubated with the total lysate at 30°C for 1 hr. The click chemistry reaction was then performed by adding the following reagents: Azide-PEG3-biotin (final concentration was 100 µM), TBTA (final concentration was 0.5 mM), $CuSO_4$ (final concentration was 1 mM), and TCEP (final concentration was 1 mM). The reaction was allowed to proceed at room temperature for 45 min, and then the total proteins were precipitated by methanol/chloroform (2.5/ 1) and washed by ice-cold methanol. After re-solubilize the protein pellets in 1.5% SDS, 1% Brij97, 100 mM NaCl and 50 mM triethanolamine, streptavidin agarose beads were added and incubated with the lysates at room temperature for 1 hr. After washing the beads three times by 0.2% SDS in PBS buffer, the beads were treated with 0.5 M hydroxylamine (pH 7.4) at room temperature for 1 hr. Then, the beads were washed three times with 0.2% SDS in PBS buffer. The beads were incubated with 6 M urea and 10 mM TCEP in PBS at 37°C for 30 min, then 400 mM iodoacetamide was added and incubated with the beads at 37°C for 30 min. After washing the beads with 2 M urea in PBS, the beads were incubated with 2 µg trypsin in 2 M urea in PBS at 37°C for 8 hr. The digestion reaction was quenched with 0.1% trifluoroacetic acid and the mixture was desalted using a Sep-Pak C18 cartridge. The lyophilized peptides were used for nano LC-MS/MS analysis. LC-MS/MS analysis was performed using LTQ-Orbitrap Elite mass spectrometer. The lyophilized peptides were dissolved in 2% acetonitrile containing 0.5% formic acid. The Orbitrap was interfaced with a Dionex UltiMate3000 MDLC system. The peptide samples were injected onto C18 RP nano column (5 µm, 75 µm × 50 cm, Magic C18, Bruker) at a flow rate of 0.3 µL/min. The gradient for HPLC condition was 5–38% acetonitrile containing 0.1% formic acid in 120 min. The Orbitrap Elite was operated in positive ion mode with spray voltage 1.6 kV and source temperature 275°C. Data-dependent acquisition (DDA) mode was used by one precursor ions MS survey scan from m/z 300 to 1800 at resolution 60,000 using FT mass analyzer, followed by up to 10 MS/MS scans at resolution 15,000 on 10 most intensive peaks. All data were acquired in Xcalibur 2.2 operation software.

## Detection of lysine fatty acylation on R-Ras2 by mass spectrometry

*Sirt6* KO MEFs stably expressing FLAG R-Ras2 were used for detecting lysine fatty acylation on R-Ras2. Cells were treated with 50 µM Alk14 for 6 hr, collected and lysed by Nonidet P-40 lysis buffer using the same method described above. 40 mg of total protein lysates were used for FLAG IP. After washing the FLAG resin three times with IP washing buffer, R-Ras2 protein was eluted by heating at 95°C for 10 min in buffer containing 1% SDS and 50 mM Tris-HCl pH 8.0. The supernatant was treated with 300 mM $NH_2OH$ pH 7.4 at 95°C for 10 min. R-Ras2 protein was then precipitated by methanol/chloroform and processed for disulfide reduction, denaturing, alkylation, and neutralization using the same method described above. The processed R-Ras2 protein was digested with 1.5 µg of trypsin in a glass vial at 37°C for 2 hr, and then desalted using Sep-Pak C18 cartridge. For

the LC-MS/MS analysis of lysine fatty acylated peptides, the same settings as SILAC experiment was applied except the LC gradient, which was 5–95% ACN with 0.1% trifluoroacetic acid from 0 to 140 min. All data were acquired in Xcalibur 2.2 operation software. RNA extraction, reverse transcription and PCR analysis of mRNA levels. Total RNAs were extracted using RNeasy Mini kit (QIAGEN). Reverse transcription was performed using SuperScript III First-Strand Synthesis kit (Invitrogen). PCR amplification was performed using Herculase II Fusion Enzyme with dNTPs Combo kit (Agilent).

## R-Ras2 localization by subcellular fractionation and confocal imaging

Subcellular fractionation and confocal imaging were performed according to reported procedures (*Huang et al., 2012*). Confocal imaging was performed on a Zeiss LSM880 confocal/multiphoton microscope.

## Statistical analysis

Data were expressed as mean ± s.d. (standard deviation, shown as error bars). Differences were examined by two-tailed Student's *t*-test between two groups; *p<0.05, **p<0.01, ***p<0.005.

## Acknowledgements

This work was supported in part by a grant from NIH/NIGMS (GM098596), NIH/NIDDK (DK107868). ODN was supported by an NIH T32 training grant (GM008500). HJ was supported by a Howard Hughes Medical Institute International Student Research Fellowship. We thank Dr. Raul Mostoslavsky at Massachusetts General Hospital for providing the *Sirt6* WT and KO MEFs, Dr. Neal M Alto at University of Texas Southwestern Medical Center for providing the plasmid of IpaJ, Dr. Wei Chen and Dr. Sheng Zhang at the Proteomic and MS Facility of Cornell University for help with the SILAC experiments. Imaging data was acquired in the Cornell BRC-Imaging Facility using the shared, NYSTEM (CO29155)- and NIH (S10OD018516)-funded Zeiss LSM880 confocal/multiphoton microscope.

## Additional information

### Funding

| Funder | Grant reference number | Author |
|---|---|---|
| National Institute of General Medical Sciences | GM098596 | Hening Lin |
| National Institute of Diabetes and Digestive and Kidney Diseases | DK107868 | Hening Lin |
| National Institute of General Medical Sciences | GM008500 | Ornealla D Nelson |
| Howard Hughes Medical Institute | International Student Research Fellowship | Hui Jing |

The funders had no role in study design, data collection and interpretation, or the decision to submit the work for publication.

### Author contributions

XZ, Conceptualization, Data curation, Investigation, Methodology, Writing—original draft; NAS, Resources, Validation, Writing—review and editing; ODN, Resources, Writing—review and editing; HJ, Validation, Investigation, Writing—review and editing; HL, Conceptualization, Supervision, Funding acquisition, Writing—original draft, Project administration, Writing—review and editing

### Author ORCIDs

Xiaoyu Zhang, http://orcid.org/0000-0002-0951-9664
Hening Lin, http://orcid.org/0000-0002-0255-2701

## Additional files

**Supplementary files**

• Supplementary file 1. Protein list of Alk12 SILAC in *Sirt6* WT and KO MEFs

• Supplementary file 2. Protein list of Alk14 SILAC in *Sirt6* WT and KO MEFs

• Supplementary file 3. Protein list of total protein SILAC in *Sirt6* WT and KO MEFs

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
