## [Decision Letter]

Thank you for submitting your work entitled "SIRT6 regulates Ras-related protein R-Ras2 by lysine defatty-acylation" for further consideration at *eLife*. Your revised article has been favorably evaluated by Tony Hunter (Senior Editor), a Reviewing Editor, and two reviewers.

Both reviewers and the Reviewing Editor found your manuscript to be interesting and its findings of sufficient general interest for the readership of *eLife*. However, some concerns were raised about interpretation of the results that we expect would require additional experimentation to satisfactorily address. In particular, both reviewers asked about the relationship between cysteine and lysine fatty acylation, in terms of potential for transacylation as well as their individual and respective contributions to R-Ras2 localization and function. There were also some questions about proteomics data processing and analysis that should be addressed. The full reviews are provided below for your benefit in preparing a revised manuscript.

Reviewer #2:

The work by Zhang et al. describes the use of a chemical probe to discover lysine fatty acylation of R-Ras2 and biochemical data that shows that SIRT6 can remove this modification. They then use a combination of shRNA and overexpression of R-Ras2 and SIRT6 mutants to show a relationship between the presence of the acylated lysine residues on R-Ras2, SIRT6 expression, and cellular viability and proliferation. The conclusions of the paper are partially supported by the presented data, in particular the evidence that SIRT6 can de-acylate R-Ras2. However, I feel that the paper would be well served by a more extensive exploration of the palmitoylation on R-Ras2 itself, as this is co-equally important to the SIRT6 activity.

Specific comments:

1) I'm curious about the relationship between palmitoylation on the cysteine residue at residue 199 and the four different lysine residues that were explored in this paper. The authors use NH_2_OH to remove the cysteine palmitoylation in Figure 2 and see some residual modification in SIRT6 knockout cells. However, they do not show a control to show that all cysteine palmitoylation is removed. The authors should blot for the loss of a control protein like H-Ras to confirm that all of the cysteine palmitoylation is removed in their conditions. These data could be readily added to a replicate of the experiment in Figure 1, which would be straight forward.

2) Furthermore, the proximity of the lysines in this study to the cysteine in R-Ras2 raises the possibility that cysteine palmitoylation occurs enzymatically and then the acyl-chain is chemically transferred to one or more lysine residues. The authors should explore this possibility by examining the palmitoylation of R-Ras2 containing a cysteine to serine mutant at residue 199.

3) Finally, the authors should explore the individual K to R mutants to determine which of the lysine residues is the major site of acylation. Maybe they are all equally important, which would be fine, but this should be directly tested. These data in comments 2 and 3 could be added to Figure 4 and would not require significant additional experimentation.

4) Many of the conclusions are based on quantification of Western blotting. There is nothing wrong with this, but some of the differences are quite small by eye. For example, the differences in phosphorylated Akt in Figure 4. The authors should show all of their replicate blots that they used for quantification as supporting figures. These replicates should be biological replicates and the authors should make this clear in their figure legends. This should be done for all quantified Western blots and not just Figure 4. The authors should already have these data for their quantification.

5) The Introduction section is quite short and would benefit from a more lengthy description of R-Ras function, and modification. More information about the sirtuins would also be helpful. Some of this is at the beginning of the Results section, but I believe this should be moved to the Introduction. Without this additional information and description, it is quite hard to place this work into the larger context of Ras and SIRT6. I think this is important for the broad readership of *eLife*.

Reviewer #3:

Zhang and Lin describe their findings that SIRT6 regulated the de-acylation of R-Ras2, a Ras-family small GTPase involved in cell growth and PI3K signaling. They begin by demonstrating the SIRT6-G60A mutant is sufficient to restore growth suppression in SIRT6 KO cells, suggesting long-chain fatty acylation of cellular substrates is directly involved in proliferation. They then carry out proteomics analysis on alk-12, alk-14, and unenriched samples. This SILAC analysis should be more clearly summarized in the supplemental tables, particularly since most of the identified proteins were not even used in the analysis (2 or more peptides, enriched by both alk-12 and alk-14, and not differentially expressed). In the current submission, the data appears to be just the Proteome Discoverer output files without any filtering. It would help to add a summary spreadsheet that shows the most critical information (peptides #, SILAC ratio, and statistics) across each experiment. They somehow arrived at a short list of candidate proteins, but how they got there is lost in the supplemental tables. They go to show that R-Ras2 knockdown reduces cell proliferation in both SIRT6-KO and WT cells, and carry out a series of elegant biochemical experiments to validate that R-Ras2 is a SIRT6 substrate that requires NAD+ and generates fatty acyl-ADPR.

Furthermore, metabolic labeling with reporter fatty acids confirmed enhanced acylation of R-Ras2. They then replaced the 4 lysines in the C-terminal HVR with 4 arginines, and showed this mutant had reduced PM localization. A recent report by Hancock (Zhou et al., 2017, Cell 168, 239-251) on K-Ras showed that the positive charge in the HVR of K-ras is not the whole story, and that K to R mutants do not fully restore proper membrane binding. Instead, the poly-Lys HVR adopts a precise conformation that achieves phospholipid binding. Thus, the differential localization could be due to decreased phospholipid interactions rather than changes in lysine acylation. The manuscript should discuss this recent publication, and adapt their conclusions (particularly in regards to why R doesn't complement K). They should also include some discussion of how their data could be extrapolated to other Ras variants, like m-Ras, K-ras, or R-Ras that use a polybasic second signal to promote membrane localization. Can the authors comment on these other Ras variants? Is this a general phenomenon of small GTPases? What is the effect of mutating the cysteine in the HVR? Does it block Lys acylation? It would also be useful to present some statistical analysis or at least show more than 1 cell from their confocal microscopy analysis. There is no mention of how many cells were analyzed.

Overall this study adds more depth to the role of Lys fatty acylation, and suggests this could be more common than initially thought. While site-specific proteomic analysis would be the most direct method to annotate these PTMs, the current study is an important step in this direction.

---

## [Author Response]

[…] Reviewer #2:

[…] 1) I'm curious about the relationship between palmitoylation on the cysteine residue at residue 199 and the four different lysine residues that were explored in this paper. The authors use NH_2_OH to remove the cysteine palmitoylation in Figure 2 and see some residual modification in SIRT6 knockout cells. However, they do not show a control to show that all cysteine palmitoylation is removed. The authors should blot for the loss of a control protein like H-Ras to confirm that all of the cysteine palmitoylation is removed in their conditions. These data could be readily added to a replicate of the experiment in Figure 1, which would be straight forward.

We agree that fluorescence labeling signal after NH_2_OH treatment may be due to the incomplete removal of cysteine palmitoylation by NH_2_OH. One experimental result that can rule out this concern is shown in Figure 2—figure supplement 2: before NH_2_OH treatment, R-Ras2 WT and 4KR showed comparable Alk14 labeling signals. NH_2_OH treatment removed almost all labeling signals from R-Ras2 4KR but not R-Ras2 WT. Considering that R-Ras2 4KR does not have lysine fatty acylation, this result suggested that NH_2_OH treatment can efficiently remove cysteine fatty acylation on R-Ras2.

Furthermore, to more convincingly demonstrate the existence of lysine fatty acylation, we have carried out mass spectrometry experiments on R-Ras2. Tandem MS/MS identified Lys194 as one of the major lysine fatty acylation site (Figure 2).

2) Furthermore, the proximity of the lysines in this study to the cysteine in R-Ras2 raises the possibility that cysteine palmitoylation occurs enzymatically and then the acyl-chain is chemically transferred to one or more lysine residues. The authors should explore this possibility by examining the palmitoylation of R-Ras2 containing a cysteine to serine mutant at residue 199.

We performed this experiment as the reviewer suggested. The new data and corresponding description are shown in Figure 2—figure supplement 2 and the main text. R-Ras2 C199S mutant can still be labeled by Alk14 (although the labeling is weaker compared WT), suggesting that cysteine fatty acylation of R-Ras2 is not essential for the occurrence of lysine fatty acylation, but it may affect lysine fatty acylation.

3) Finally, the authors should explore the individual K to R mutants to determine which of the lysine residues is the major site of acylation. Maybe they are all equally important, which would be fine, but this should be directly tested. These data in comments 2 and 3 could be added to Figure 4 and would not require significant additional experimentation.

We performed this experiment as the reviewer suggested. The new data is shown in Figure 2. Interestingly, the NH_2_OH resistant labeling of all the single mutants was similar to that of WT, suggesting that K192, K194, K196, and K197 are likely to be fatty acylated redundantly. In the revised manuscript, we also performed mass spectrometry and identified Lys194 as the fatty acylation site on WT R-Ras2 protein (Figure 2). The data suggests that each signal lysine can be modified redundantly, but on WT R-Ras2 protein, Lys194 may be the major fatty acylation site.

4) Many of the conclusions are based on quantification of Western blotting. There is nothing wrong with this, but some of the differences are quite small by eye. For example, the differences in phosphorylated Akt in Figure 4. The authors should show all of their replicate blots that they used for quantification as supporting figures. These replicates should be biological replicates and the authors should make this clear in their figure legends. This should be done for all quantified Western blots and not just Figure 4. The authors should already have these data for their quantification.

We added all of the replicate western blots that were used for quantification in Figure 4—figure supplement 2.

5) The Introduction section is quite short and would benefit from a more lengthy description of R-Ras function, and modification. More information about the sirtuins would also be helpful. Some of this is at the beginning of the Results section, but I believe this should be moved to the Introduction. Without this additional information and description, it is quite hard to place this work into the larger context of Ras and SIRT6. I think this is important for the broad readership of eLife.

As the reviewer suggested, we moved the introduction of SIRT6 from the Results section to the Introduction section. We also added more description of R-Ras2 and sirtuins in the Introduction section.

Reviewer #3:

Zhang and Lin describe their findings that SIRT6 regulated the de-acylation of R-Ras2, a Ras-family small GTPase involved in cell growth and PI3K signaling. They begin by demonstrating the SIRT6-G60A mutant is sufficient to restore growth suppression in SIRT6 KO cells, suggesting long-chain fatty acylation of cellular substrates is directly involved in proliferation. They then carry out proteomics analysis on alk-12, alk-14, and unenriched samples. This SILAC analysis should be more clearly summarized in the supplemental tables, particularly since most of the identified proteins were not even used in the analysis (2 or more peptides, enriched by both alk-12 and alk-14, and not differentially expressed). In the current submission, the data appears to be just the Proteome Discoverer output files without any filtering. It would help to add a summary spreadsheet that shows the most critical information (peptides #, SILAC ratio, and statistics) across each experiment. They somehow arrived at a short list of candidate proteins, but how they got there is lost in the supplemental tables. They go to show that R-Ras2 knockdown reduces cell proliferation in both SIRT6-KO and WT cells, and carry out a series of elegant biochemical experiments to validate that R-Ras2 is a SIRT6 substrate that requires NAD+ and generates fatty acyl-ADPR.

In the revised manuscript, we added a new figure (Figure 1) showing how the SILAC data was analyzed. We also included the peptide #, SILAC ratio, and statistics in the revised Figure 1.

Furthermore, metabolic labeling with reporter fatty acids confirmed enhanced acylation of R-Ras2. They then replaced the 4 lysines in the C-terminal HVR with 4 arginines, and showed this mutant had reduced PM localization. A recent report by Hancock (Zhou et al., 2017, Cell 168, 239-251) on K-Ras showed that the positive charge in the HVR of K-ras is not the whole story, and that K to R mutants do not fully restore proper membrane binding. Instead, the poly-Lys HVR adopts a precise conformation that achieves phospholipid binding. Thus, the differential localization could be due to decreased phospholipid interactions rather than changes in lysine acylation. The manuscript should discuss this recent publication, and adapt their conclusions (particularly in regards to why R doesn't complement K).

We would like to thank the reviewer for pointing out this recent paper, which emphasize the distinct role of lysine from that of arginine. We added the following discussion in the manuscript:

“In a recent paper, the lysine residues at the C-terminal of K-Ras4B have been shown to play important roles for phospholipid binding and K-Ras4B signal output (Rodriguez-Viciana, Sabatier and McCormick, 2004). […] However, we could not completely rule out that the lysine residues at the C-terminal of R-Ras2 may also function in phospholipid binding and thus affect its signaling output, in addition to being regulated by lysine fatty acylation and SIRT6.”

They should also include some discussion of how their data could be extrapolated to other Ras variants, like m-Ras, K-ras, or R-Ras that use a polybasic second signal to promote membrane localization. Can the authors comment on these other Ras variants?

We added more discussions about other Ras proteins. We had the same speculation that other Ras variants possessing polybasic sequence might be regulated by lysine fatty acylation. Indeed, we have preliminary data suggesting several other Ras variants are fatty acylated on lysine residues at the C-terminal HVR region. We have two manuscripts in preparation focusing on the functional study of lysine fatty acylation of these Ras proteins.

Is this a general phenomenon of small GTPases? What is the effect of mutating the cysteine in the HVR? Does it block Lys acylation?

We believe this is a general phenomenon that occurs to many small GTPases, as mentioned above. Mutating the palmitoylated cysteine does not block lysine acylation (Figure 2—figure supplement 2), but slightly decreased lysine acylation.

It would also be useful to present some statistical analysis or at least show more than 1 cell from their confocal microscopy analysis.

We have added this information in the revised manuscript (Figure legends 2J and 3C). We also included the other images used in supplementary figures (Figure 2—figure supplement 4 and Figure 3—figure supplement 1).